**Data Availability Statement:** Data cannot be shared publicly because Ethical restrictions have been imposed on the data underlying this study by

# Practice assistants´ perceived mental workload: A cross-sectional study with 550 German participants addressing work content, stressors, resources, and organizational structure

Jan Hoffmann[1¤a]*, Christine Kersting[2¤b], Birgitta Weltermann[1]

**1** Institute of General Practice and Family Medicine, University Hospital of Bonn, Bonn, Germany, **2** Institute for General Medicine, University Hospital Essen, University of Duisburg-Essen, Essen, Germany

¤a Current address: Institute of Medical Sociology, Health Services Research and Rehabilitation Science, University of Cologne, Cologne, Germany
¤b Current address: Institute of General Practice and Interprofessional Care, Faculty of Health/School of Medicine, Witten/Herdecke University, Witten, Germany

* jan.hoffmann@uk-koeln.de

## Abstract

### Introduction

Practice assistants represent a highly relevant occupational group in Germany and one of the most popular training professions in Germany. Despite this, most research in the health care sector has focused on secondary care settings, but has not addressed practice assistants in primary care. Knowledge about practice assistants' workplace-related stressors and resources is particularly scarce. This cross-sectional study addresses the mental workload of practice assistants working in primary care practices.

### Methods

Practice assistants from a network of 185 German primary care practices were invited to participate in this cross-sectional study. The standardized 'Short Questionnaire for Workplace Analysis' (German: Kurzfragebogen zur Arbeitsanalyse) was used to assess practice assistants´ mental workload. It addresses eleven workplace factors in 26 items: versatility, completeness of task, scope of action, social support, cooperation, qualitative work demands, quantitative work demands, work disruptions, workplace environment, information and participation, and benefits. Sociodemographic and work characteristics were also obtained. A descriptive analysis was performed for sociodemographic data and "Short Questionnaire for Workplace Analysis" factors. The one-sided t-test and Cohen´s d were calculated for a comparison with data from 23 professional groups (n = 8,121).

### Results

A total of 550 practice assistants from 130 practices participated. The majority of practice assistants was female (99.3%) and worked full-time (66.5%) in group practices (50.6%).

the Ethics Committee of the Medical Faculty of the University of Duisburg-Essen in order to protect participant confidentiality. However, requests for an ethically compliant dataset may be made to the Ethics Committee of the Medical Faculty of the Universtiy of Duisburg-Essen (ethikkommission@uk-essen.de).

**Funding:** Funding for this study was provided by the Ministry of Culture and Science, North-Rhine Westphalia, Germany, formerly the Ministry of Innovation, Science and Research, North-Rhine Westphalia, Germany. https://www.mkw.nrw/. The funders had no role in study design, data collection and analysis, decision to publish, or preparation of the manuscript.

**Competing interests:** The authors have declared that no competing interests exist.

**Abbreviations:** CI, confidence interval; GP, general practitioner; KFZA, Kurzfragebogen zur Arbeitsanalyse (English: Short Questionnaire for Workplace Analysis); PrA, practice assistant; SD, standard deviation.

Compared to the other professional groups, practice assistants reported higher values for the factor social support (4.0 versus 3.7 [d 0.44; p<0.001]), information and participation (3.6 versus 3.3 [d 0.38; p<0.001] as well as work disruptions (2.7 vs. 2.4 [d 0.42; p<0.001]), while practice assistants showed lower values regarding scope of action (3.4 versus 3.8 [d 0.43; p<0.001]).

## Conclusions

Our study identified social support and participation within primary care practices as protective factors for mental workload, while work disruptions and scope of action were perceived as stressors.

## Introduction

Practice assistants (PrAs) represent the largest group of employees in the German outpatient health care sector [1] and the second most popular training profession among German women [2]. However, little is known about how PrAs perceive their work conditions. More specifically, data on the relationship between work and psychological stress in PrAs are lacking. While psychosocial assessment studies of health personnel in secondary care have been performed [3–6], only few have addressed this issue in PrAs in German primary care [1, 7, 8]. Therefore, it is important to further investigate PrAs' perceived level of psychological stress, as psychological strain may not only threaten PrAs' health with potentially tremendous economic costs, but may also impair high-quality patient care [9].

In recent years, increasing attention has been devoted to employees' mental health. A systematic review by Theorell et al. highlighted that job strain has an impact on the development of depressive symptoms [10]. Also, the socio-economic implications are increasingly evident: preceded only by musculoskeletal diseases, mental health conditions rank second with 16.7% of all sick leaves among German employees [11] and caused a damage of 21.7 billion Euros gross added value in 2017 [11].

The stress-strain model developed by Rohmert and Rutenfranz in 1975 differentiates between the terms 'psychological stress' and 'psychological strain'. 'Psychological stress' describes all external factors that influence one's psychological well-being. When referring to psychological stress in a work environment, the term 'mental workload´ refers to employees´ exposure to individual work demands and the environment at work [12]. However, the term does not necessarily have a negative connotation [13]. 'Psychological strain' can be understood as an individual´s response to psychological stress. Thus, the same level of psychological stress may elicit a different level of psychological strain depending on an employee´s coping strategy and constitution [14]. A well-balanced amount of psychological strain can lead to a healthy and productive workflow [12], while an extreme level of psychological strain may threaten employees' health. Studies have shown a negative association between high levels of psychological strain and mental illness [15, 16].

Since 2014, the German Safety and Health at Work Act (ArbSchG) obliges employers to perform a general risk assessment of their employees' work conditions [17]. Assessing the mental workload (a so-called 'psychosocial risk assessment´) is part of this risk assessment. Based on the results, employers must take countermeasures as necessary to enhance their employees' health [18]. Due to differences in work demands, work hazards, and work environments across professions there is no gold standard that defines what instrument should be

used for the psychosocial risk assessment. While different instruments exist [19], the so-called Kurzfragebogen zur Arbeitsanalyse (KFZA; English: Short Questionnaire for Workplace Analysis), a questionnaire addressing perceived workload, is widely used across professions [20]. Data from more than 8,000 participants from 23 professions are available [8].

The aims of this cross-sectional study are threefold: i) to assess the mental workload of PrAs working in German primary care practices, ii) to identify resources and stressors, and iii) to compare results with aggregated data from 23 different professions.

## Material and methods

### Study design and recruitment of participants

The psychosocial assessment of PrAs reported in this paper was obtained as part of a larger cross-sectional study investigating multiple aspects of stress in primary care practices. Details of the study are reported elsewhere [21, 22]. Briefly, general practitioners (GPs) and PrAs of the 185 general medicine practices of the practice network of the Institute for General Medicine, University Hospital Essen, Essen, Germany, were asked to participate in the study. The practices were located in urban and rural regions of North Rhine-Westphalia (Western Germany) with an average distance of 30 km (range: 2±180 km) to the Institute. In a prior study it was shown that the practices affiliated with the network are representative for German primary care practices [23]. Practices had been invited by mail and contacted by phone for further recruitment. Those refusing to participate were asked to answer a short questionnaire on practice characteristics and to provide reasons for non-participation. Data were collected between April and September 2014 during on-site visits. Within each practice, all GPs (practice owners and employed physicians) and PrAs including medical secretaries and PrA trainees were eligible for participation and received the study documents. The study documents comprised a study information sheet, an informed consent form to be completed by all participants, and a set of questionnaires which included sociodemographic questions and the KFZA analyzed in this paper. To ensure data protection, participants were asked to seal the completed questionnaire in an envelope. As an incentive, practice teams received a department store chain voucher of 5 euros per person, irrespective of the participation of individual team members. In addition, the dataset contained information about the practices´ location from the practice network´s database and matched with public regional data for the population size in 2012 (www.it.nrw.de). This paper follows the STROBE recommendations for reporting cross-sectional studies [24].

Ethical approval had been obtained from the Ethics Committee of the Medical Faculty of the University of Duisburg-Essen (reference number: 13-5536-BO, date of approval: 24/11/2014). All participants received written information and signed informed consent forms.

### Study instrument to assess mental workload

The KFZA was developed by Prümper et al. in 1995 and is as a widely accepted screening tool for psychological stress at the workplace [25]. The questionnaire is a standardized instrument with closed questions. It is completed by the employees themselves and thus provides a subjective view of each individual's perception of the work environment. According to DIN EN ISO 10075 "Ergonomic principles related to mental workload", the instrument is categorized as a "precision level 2 process for overview purposes" [26]. The instrument is listed in the toolbox for "Instruments for recording mental loads" of the Federal Institute for Occupational Safety and Health and covers multiple aspects of the work environment [27]. It includes four dimensions: *work content*, *resources*, *stressors*, *and organizational culture*. Dimensions consist of 11 factors which are derived from 26 single items with answer options on a Likert scale

ranging from 1 (does not apply at all) to 5 (is completely true). *Work content* contains two factors (versatility, completeness of task) and five single items (learning new skills, use of knowledge, skills and ability, variety of tasks, visibility of task accomplishment, completeness of product). *Resources* contains three factors (scope of action, social support, cooperation) and nine single items (influence on sequence of activities, influence on work content, influence on workload and procedures, social support by co-workers, social support by supervisors, social cohesion within the department, necessity of cooperation, opportunity for social exchange with co-workers, feedback from supervisors and co-workers). *Stressors* contains four factors (qualitative work demands, quantitative work demands, work disruptions, workplace environment) and eight single items (excessive complexity of tasks, excessive demands on concentration, frequent work under time pressure, too much work to do, lack of information, work materials or equipment, interruptions of workflow, unfavorable physicochemical conditions, insufficient workspace and equipment). *Organizational culture* contains two factors (information and participation, benefits) and four single items (information about organizational developments, consideration of employee input, continuous education, opportunities for advancement). The dimensions job content, resources, and organizational culture represent positive aspects, and high scores are considered positive. High scores in the stressors dimension are considered negative work aspects.

Given the time constraints in primary care practices, the KFZA was deemed suitable as it takes only 10 minutes to complete. Also, data from more than 8,000 participants from 23 other professional groups are available for comparison [25]. The questionnaire can be applied throughout all professions and workspaces and is readily available for academic use [28].

## Comparative data from 23 professional groups

In 2000, the Employers' Liability Insurance Association for Medical Services and Welfare Work (BGW) in cooperation with the German Employees' Health Insurance (DAK) conducted a cross-sectional study to measure stress at work [8]. A purposive sample of 27,584 employees from 23 professional groups was selected from the BGW and DAK register: physicians, assistant pharmacists, pharmacists, office workers, teacher, hairdressers, pest controllers, alternative practitioners, unskilled laborers, kindergarten teachers, chefs, nurses, masseurs, medical laboratory technicians, porters, facility cleaners, social workers, PrAs, veterinarians, care workers for persons at risk, employees of dialysis centers, and employees of workshops for the disabled. A total of 8,121 employees participated in the study in the context of a project called 'Prevention of work-related health hazards'. The KFZA was used within the scope of the study. We performed two comparative analyses using published data of the survey: first, we compared KFZA results from the study of the 23 professional groups with results from our population. Second, we compared the results for the subpopulation of PrAs from the study with results from our population. The latter comparison is particularly interesting, as it provides a longitudinal approach (data from 2000 and 2014) in a situation where the vocational training was meanwhile been revised and PrAs in Germany are professionalizing.

## Data analysis

The analysis was performed using IBM SPSS Statistics for Windows, Version 25 (Armonk, NY: IBM Corp.). Data of all PrAs were analyzed. Non-plausible values were recoded as missing values. Missing data were managed by reporting valid percentages only.

Sociodemographic and work-related characteristics were analyzed descriptively. The mean, standard deviation (SD), median, and range are reported for metric sociodemographic and work variables. The practices' population size was categorized into rural, small, medium-sized,

and big cities following categorization schemes of the Federal Institute for Research on Building, Urban Affairs and Spatial Development (rural $\leq$ 4,999 inhabitants, small city 5000–19,999, medium-sized city 20,000–99,9999, big city $\geq$ 100,000).

Following Prümper et al., the results of the KFZA were evaluated by computing mean values on a factor level [25, 29]: As a first overview, positive items <3 and negative items >3 are interpreted as high levels of psychological stress and indicate a need for more detailed analyses. In addition, the comparison with data from other professional groups or from the same professional group provides information on how to set a benchmark against other results [29]. Differences between the means of our population and the comparative population were analyzed using a one-sided t-test (95% significance level; 0.05 = alpha). Additionally, Cohen´s d was calculated to estimate the effect size. 95% confidence intervals (CI) were calculated for factors of the 2014 PrA population. Power calculations were performed using the software G-Power 3.1 to determine the appropriateness of sample sizes used in the group comparisons.

## Results

### Study characteristics

550 PrAs participated in the study (response rate 70.3%; n = 130 practices). The sociodemographic characteristics of the participants are presented in Table 1. PrAs had a mean age of 37.97 years (SD: 12.63), with 99.27% of PrAs being female. The majority of PrAs was married (50.64%), worked full-time (66.48%) on a permanent contract (89.25%) with a median work experience of 18 years (range: 0–49 years). Most (62.59%) PrAs worked 20–39 hours a week, while 25.37% of PrAs worked more than 39 hours. Most PrAs (93.87%) had completed a three-year vocational training as "Medizinische Fachangestellte" or "Arzthelferin" which combines practical training (3 days per week) and vocational training (2 days per week). Six percent had other backgrounds (i.e.: secretary, practice aid, other practice employee or a vocational training stated in the further comments section labeled as "other"). Almost all PrAs had completed some sort of additional training: 19.23% of PrAs had completed additional training as VERAHs (106 hours of theoretical and 94 hours of practical training) or EVAs (170 to 220 hours of theoretical training and 20 to 50 hours of practical training depending on prior work experience) that allows PrAs to perform additional tasks (e.g.: home visits). On average, PrAs worked in practices with 2.96 (SD 2.15) physicians and 7.73 (SD 7.64) PrAs. Half of the practices (50.64%) were group practices. The smallest proportion of PrAs worked in practices with a low patient load per quarter (5.59%, 501–1000 patients per quarter), while the largest proportion of PrAs worked in practices with a high patient load per quarter (27.93%, >3001 patients per quarter). PrAs' work setting characteristics are presented in Table 2.

### Comparison of practice assistants with other professional groups (comparative data)

The power calculation revealed that the sample sizes compared (n = 550 versus n = 8.121) were sufficient to achieve 80% power to detect small effect sizes of d = 0.12. In the case of greater differences, the power achieved was even higher.

Table 3 shows the results of the KFZA analysis for PrAs and for the comparative population. For a first overview of only results from our study population, the calculation of mean values for the factor-level analysis yielded a critical score for the factor benefits (2.86 [SD 1.05]). In contrast, social support showed the highest positive factor (4.05 [SD 0.79]).

As illustrated in Fig 1, the comparison of our results with data from Nolting et al. [8] revealed statistically significant differences (p < 0.05) for the following factors: versatility (3.6

**Table 1. Practice assistants´ sociodemographic and professional training characteristics (n = 550).**

| Variable | Total (n = 550) | 100%* |
|---|---|---|
| **Age (n = 550, years)** | | |
| Mean (SD) | 37.97 | (12.63) |
| Median (min-max) | 38 | (16–71) |
| **Gender (n = 548)** | | |
| Female | 542 | 99.27 |
| Male | 4 | 0.73 |
| **Marital status (n = 547)** | | |
| Single | 218 | 39.85 |
| Married | 277 | 50.64 |
| Divorced | 45 | 8.23 |
| Widowed | 7 | 1.28 |
| **Status of employment (n = 534)** | | |
| Full-time | 355 | 66.48 |
| Part-time | 179 | 33.52 |
| **Mode of employment (n = 521)** | | |
| Fixed-term | 56 | 10.75 |
| Permanent | 465 | 89.25 |
| **Working hours per week (n = 541)** | | |
| 0–19 | 65 | 12.04 |
| 20–39 | 338 | 62.59 |
| 40–59 | 127 | 23.52 |
| >60 | 10 | 1.85 |
| **Work experience (n = 540, years)** | | |
| Mean (SD) | 18.74 | (12.46) |
| Median (Min-Max) | 18 | (0–49) |
| **PrA in training** | | |
| Yes | 49 | 8.94 |
| No | 499 | 91.06 |
| **Year of training (n = 47)** | | |
| First year | 16 | 34.04 |
| Second year | 19 | 40.43 |
| Third year | 12 | 25.53 |
| **Vocational training [1] (n = 522)** | | |
| Practice assistants | 490 | 93.87 |
| Secretary | 12 | 2.30 |
| Practice aid[2] | 6 | 1.15 |
| Other practice employees[2] | 16 | 3.07 |
| Other | 75 | 14.37 |
| **Additional training (n = 130)** | | |
| VERAH | 14 | 10.77 |
| EVA | 3 | 2.31 |
| VERAH/EVA + other | 8 | 6.15 |
| Other | 105 | 80.77 |

[1] multiple answers possible

[2] no vocational training.

**Table 2. Practice assistants' work setting characteristics (n = 550).**

| Variable | Total (n = 550) | 100% |
|---|---|---|
| **Type of practice (n = 545)** | | |
| Solo practice | 147 | 26.97 |
| Group practice | 276 | 50.64 |
| Others | 122 | 22.39 |
| **Number of patients per quarter (n = 537)** | | |
| 501–1000 | 30 | 5.59 |
| 1001–1500 | 116 | 21.60 |
| 1501–2000 | 100 | 18.62 |
| 2001–2500 | 79 | 14.71 |
| 2501–3000 | 62 | 11.55 |
| >3001 | 150 | 27.93 |
| **Location of practice[1] (n = 532)** | | |
| Small city | 33 | 6.20 |
| Medium-sized city | 128 | 24.06 |
| Big city | 371 | 69.74 |
| **Number of physicians in practice (n = 545, physicians)** | | |
| Mean (SD) | *2.96* | *(2.15)* |
| Median (Min-Max) | *2* | *(1–10)* |
| **Number of PrAs in practice (n = 517, PrAs)** | | |
| Mean (SD) | *7.73* | *(7.64)* |
| Median (Min-Max) | *5* | *(0–35)* |

[1] based on 2012 number of inhabitants.

vs. 3.8), completeness of task (3.5 vs. 3.6), scope of action (3.4 vs. 3.8), social support (4.0 vs. 3.7), cooperation (3.6 vs. 3.4), qualitative work demands (2.2 vs. 2.1), work disruptions (2.7 vs. 2.4), information and participation (3.6 vs. 3.3), and benefits (2.9 vs. 2.4). The two factors workplace environment (2.2 vs. 2.2) and quantitative work demands (2.9 vs. 3.0) were found to be non-significant.

**Table 3. KFZA results from our study of practice assistants (n = 550) in comparison with comparative data from 23 professional groups (n = 8.121).**

| Work aspects | KFZA factor | Our study Mean score (PrAs) | 95% CI | Comparison: Mean score (Nolting et al.) | Cohen´s d | P-value[**] |
|---|---|---|---|---|---|---|
| **Job content[1]** | **Versatility** | **3.6** | 3.58–3.70 | 3.8 | 0.23 | < 0.001 |
| | **Completeness of task** | **3.5** | 3.41–3.57 | 3.6 | 0.12 | 0.0045 |
| **Resources[1]** | **Scope of action** | **3.4** | 3.37–3.49 | 3.8 | 0.43 | < 0.001 |
| | **Social support** | **4.0** | 3.98–4.12 | 3.7 | 0.44 | < 0.001 |
| | **Cooperation** | **3.6** | 3.53–3.66 | 3.4 | 0.24 | < 0.001 |
| **Stressors[2]** | **Qualitative work demands** | **2.2** | 2.14–2.29 | 2.1 | 0.13 | 0.0025 |
| | **Quantitative work demands** | **2.9** | 2.83–3.01 | 3.0 | 0.07 | 0.0797 |
| | **Work disruptions** | **2.7** | 2.67–2.81 | 2.4 | 0.41 | < 0.001 |
| | **Workplace environment** | **2.2** | 2.13–2.30 | 2.2 | 0.02 | 0.7109 |
| **Organizational culture[1]** | **Information and participation** | **3.6** | 3.57–3.73 | 3.3 | 0.38 | < 0.001 |
| | **Benefits** | **2.9***  | 2.77–2.94 | 2.4* | 0.43 | < 0.001 |

[1] High scores (>3) are considered positive

[2] high scores (>3) are considered negative

* critical values ** based on a one-sided t-test comparing mean values of PrAs and Nolting et al. on a 95% significance level.

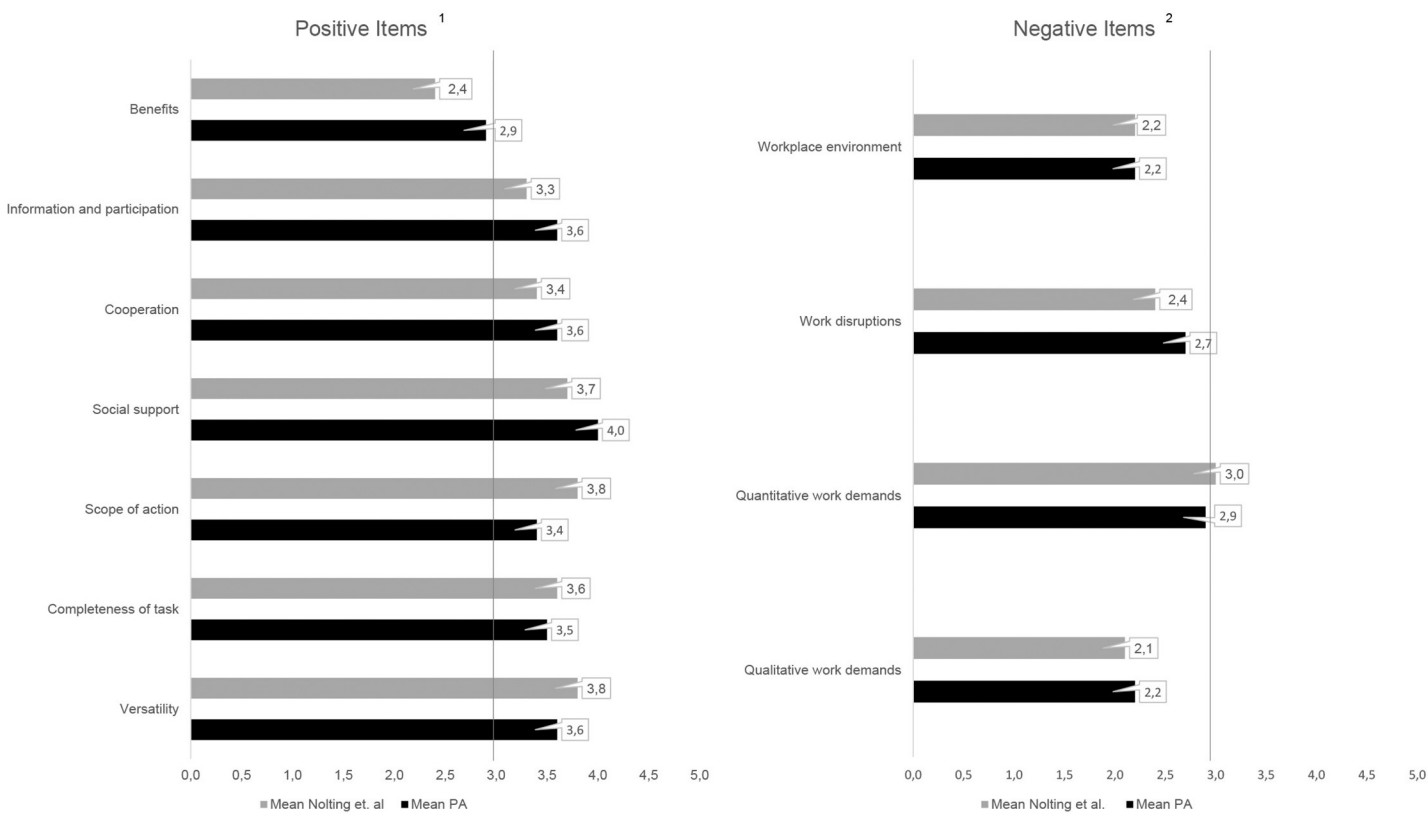

**Fig 1. KFZA results on a factor level divided into resources and stressors in comparison with comparative data from Nolting et al.** [1] High scores (>3) are considered positive, [2] high scores (>3) are considered negative.

Effect size showed the strongest difference for the factors social support (4.0 vs 3.7 [d 0.44]), scope of action (3.4 vs. 3.8 [d 0.43]), and benefits (2.9 vs. 2.4 [d 0.43]). The scores for social support and benefits were higher in the PrA population than in the comparative group, whereas scope of action yielded lower scores. The factor benefits, on the other hand, was critically low in both populations. The difference in work disruptions (2.7 vs. 2.4 [d 0.41]) presented a moderate effect size. The score for work disruptions was higher in the PrA population compared to the population from Nolting et al. [8].

## Comparison of practice assistants from 2000 and 2014

The power calculation revealed that the sample sizes compared (n = 550 versus n = 324) were sufficient to achieve 80% power to detect small effect sizes of d = 0.2. In the case of greater differences, the power achieved was even higher.

Table 4 shows the comparison between PrAs in our study population (from 2014) and the comparative study population (from 2000). The comparison yielded statistically significant differences (p < 0.05) for the factors completeness of task (3.5 vs. 3.2), social support (4.0 vs. 3.9), cooperation (3.6 vs. 3.5), qualitative work demands (2.2 vs. 2.0), quantitative work demands (2.9 vs. 2.8), work disruptions (2.7 vs. 2.5), workplace environment (2.2 vs. 2.0), information and participation (3.6 vs. 3.5), and benefits (2.9 vs 2.2).

Effect size showed no effect for versatility (d 0.05), scope of action (d 0.01), social support (d 0.19), cooperation (d 0.13), quantitative work demands (d 0.12), as well as information and participation (d 0.16). A small effect size was shown for completeness of task (d 0.32),

**Table 4. KFZA factor-level comparison of PrAs from our study (n = 550; year 2014) and PrAs from Nolting et al. (n = 324; year 2000).**

| Work aspects | KFZA factor | Our study<br>Mean score (PrAs) | 95% CI | PrAs' results from 2000<br>Mean score (PrAs; Nolting et al.) | Cohen´s d | P-value |
|---|---|---|---|---|---|---|
| Job content[1] | Versatility | 3.6 | 3.58–3.70 | 3.6 | 0.05 | 0.238 |
| | Completeness of task | 3.5 | 3.41–3.57 | 3.2 | 0.32 | < 0.001 |
| Resources[1] | Scope of action | 3.4 | 3.37–3.49 | 3.4 | 0.01 | 0.765 |
| | Social support | 4.0 | 3.98–4.12 | 3.9 | 0.19 | < 0.001 |
| | Cooperation | 3.6 | 3.53–3.66 | 3.5 | 0.13 | 0.006 |
| Stressors[2] | Qualitative work demands | 2.2 | 2.14–2.29 | 2.0 | 0.25 | < 0.001 |
| | Quantitative work demands | 2.9 | 2.83–3.01 | 2.8 | 0.12 | 0.007 |
| | Work disruptions | 2.7 | 2.67–2.81 | 2.5 | 0.29 | < 0.001 |
| | Workplace environment | 2.2 | 2.13–2.30 | 2.0 | 0.21 | < 0.001 |
| Organizational culture[1] | Information and participation | 3.6 | 3.57–3.73 | 3.5 | 0.16 | 0.002 |
| | Benefits | 2.9* | 2.77–2.94 | 2.2* | 0.62 | < 0.001 |

[1] High scores (>3) are considered positive

[2] high scores (>3) are considered negative

* critical values ** based on a one-sided t-test comparing mean values of PrAs and Nolting et al. on a 95% significance level.

qualitative work demands (d 0.25), work disruptions (d 0.29), and workplace environment (d 0.21). The difference in the factor benefits presented a moderate effect size (d 0.62).

## Discussion

Our study identified social support within primary care practices as a resource and a protective factor for mental workload among PrAs, while the lack of benefits at work was perceived as a stressor.

When comparing data on PrAs with the aggregated data of other professional groups, we were able to perform a more informative analysis yielding slightly different results. Scope of action and work disruptions showed the largest negative difference and the strongest effect size, whereas social support and benefits showed the largest positive difference and the strongest effect size. Interestingly, when comparing with other professional groups, the factor benefits that was identified as a stressor in the single evaluation turned out to be a resource. Since the scores are rather low in both samples, lack of benefits at work might be a general problem, while PrAs might experience more benefits at work than other professional groups. PrAs in general practices tend to be responsible for a wide range of tasks in different workplaces throughout the practices, as they are the first point of contact for patients with unexpected events occurring on a regular basis [1]. This job profile may explain the high scores for work disruptions. Although PrAs are responsible for a wide range of tasks, GPs remain the decision makers, resulting in a setting-immanent limited scope of action for PrAs.

The comparison between the PrA groups from 2000 to 2014 revealed significant differences for most factors, but small effect sizes. The factor benefits showed a moderate effect size in favor of the 2014 study population. All factors, positive factors and negative factors alike, were slightly higher in our population of PrAs compared to the 2000 PrA population from Nolting et al. The increase in benefits at work and completeness of task from 2000 to 2014 may be explained by the further training opportunities for PrAs that were introduced during that time period (i.e., VERAH, EVA). Among other changes, these trainings have enabled PrAs to carry out more complex work processes autonomously (e.g.: patient education on diabetes). Additionally, they are rewarded with a better salary. Both may be signs of professionalization. In a

recent study by Vu-Eickmann et al., PrAs reported a high patient volume, which in addition to handling many tasks at once may explain the high score for work disruptions [1].

Social support is an important resource and can positively influence job satisfaction, as shown in a recent study with Portuguese nursing staff [30]. Job satisfaction was again shown to positively correlate with patient satisfaction [31]. A systematic review yielded a similar result linking social support with staff well-being in emergency departments [32]. In contrast, studies have shown that negative work aspect (i.e.: lack of benefits, limited scope of action) cause psychological strain and can lead to a higher turnover rate and depressive symptoms [10, 33].

In agreement with three other studies on this topic, we showed that PrAs in primary care practices receive high social support and have a rather limited scope of action and still insufficient benefits at work [1, 7, 8].

## Strengths and limitations

It is a strength of our study that it was based on a data set with a large number of participants (550 PrAs). Also, prior analyses had shown that the practice network from which this sample was taken is representative for German primary care practices [23]. Each participant received an incentive in the form of a 5-Euro voucher to avoid a selection bias by selecting only highly motivated PrAs. As the network is located in a rather densely populated area, our results may overrepresent PrAs working in urban areas. The KFZA proved to be a cost-effective screening tool to gain first insights into employees' psychological stressors and resources. To our knowledge this is the first study comparing PrAs' data from a psychological risk assessment in primary care with a large sample from other professions.

In our study we were only able to assess the current situation and not the state desired by PrAs, which could have provided even more insights. The comparison with data from 23 professional groups was limited as only aggregated mean results were available without standard deviations. Due to this, we were unable to calculate confidence intervals for both populations. A strength of our study is the comparison of the results of the 2000 with the 2014 study from the same professional group. However, the PrA populations were not identical, and caution is advised when interpreting the results.

## Conclusions

Mental well-being has a tremendous impact on preserving a healthy and productive workforce. Therefore, our goal must be to first identify risk factors for mental well-being at work and put them into perspective with other occupations, which we aimed to do in this study. Second, we need to develop measures to tackle risk factors for psychological strain at work and enhance protective factors such as social support, scope of action, benefits at work, and cooperation. Last, measures need to be evaluated and implemented in the everyday working life of PrAs.

## Acknowledgments

We thank the Institute for General Medicine, University Hospital Essen, for supporting the conceptualization of the questionnaire, the data collection, and the provision of the data for this analysis.

## Author Contributions

**Conceptualization:** Christine Kersting, Birgitta Weltermann.

**Data curation:** Jan Hoffmann, Christine Kersting.

**Formal analysis:** Jan Hoffmann.

**Funding acquisition:** Birgitta Weltermann.

**Investigation:** Christine Kersting, Birgitta Weltermann.

**Methodology:** Jan Hoffmann, Birgitta Weltermann.

**Project administration:** Birgitta Weltermann.

**Validation:** Jan Hoffmann.

**Visualization:** Jan Hoffmann.

**Writing – original draft:** Jan Hoffmann.

**Writing – review & editing:** Christine Kersting, Birgitta Weltermann.

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
