## [Decision Letter · Decision Letter 0]

7 Jul 2020

PONE-D-20-07803

Practice assistants´ perceived mental work load: a cross-sectional study with 550 German participants addressing work content, stressors, resources, and organizational structure

PLOS ONE

Dear Dr. Hoffmann,

Thank you for submitting your manuscript to PLOS ONE. After careful consideration, we feel that it has merit but does not fully meet PLOS ONE’s publication criteria as it currently stands. Therefore, we invite you to submit a revised version of the manuscript that addresses the points raised during the review process.

Your manuscript has been assessed by an acknowledged expert in the field covered by the study. You will find below (see the appended comments) that the valuation of the paper contents has been really positive (most of the comments are relatively minor; please also see the sanitized copy of the paper attached to the review report). Nevertheless, and apart from the aforementioned issues, the English writing of the paper needs serious revisions ir order to consider the paper for acceptance in PLOS ONE. My suggestion to the authors is to summit is to a proofread carried out by a professional (or native) translator.

We look forward to receiving your revised manuscript.

Kind regards,

Sergio A. Useche, Ph.D.

Academic Editor

PLOS ONE

Journal Requirements:

4. Please ensure that you refer to Figure 1 in your text as, if accepted, production will need this reference to link the reader to the figure.

5. We note you have included a table to which you do not refer in the text of your manuscript. Please ensure that you refer to Table 4 in your text; if accepted, production will need this reference to link the reader to the Table.

Reviewers' comments:

Reviewer's Responses to Questions

**Comments to the Author**

1. Is the manuscript technically sound, and do the data support the conclusions?

Reviewer #1: Yes

2. Has the statistical analysis been performed appropriately and rigorously? 

Reviewer #1: Yes

3. Have the authors made all data underlying the findings in their manuscript fully available?

Reviewer #1: No

4. Is the manuscript presented in an intelligible fashion and written in standard English?

Reviewer #1: No

5. Review Comments to the Author

Reviewer #1: 1. The manuscript is full of typography errors; punctuations,

2. Language is main problem

3. Not consistent throughout the document

4. Don’t use abbreviation in the abstract part

5. In the background part there are incomplete sentences

6. In tables the decimal places should be consistent

7. In the table reporting missing value is not necessary

6. PLOS authors have the option to publish the peer review history of their article (what does this mean?). If published, this will include your full peer review and any attached files.

Reviewer #1: No

---

## [Author Response · Author response to Decision Letter 0]

3 Aug 2020

Journal Requirements:

Answer: Our manuscript now meets all style requirements.

Answer: The manuscript was proofread by a professional medical translator (Sarah Chalmers; https://www.medi-translate.com/) 

Answer: The data cannot be shared publicly because of ethical restrictions and data protection issues as our dataset includes potentially identifying information. 

4. Please ensure that you refer to Figure 1 in your text as, if accepted, production will need this reference to link the reader to the figure. 

Answer: A reference to Figure 1 is now included (line 208).

 “As illustrated in Fig 1, the comparison of our results with data from Nolting et al. [8] revealed statistically significant differences (p < 0.05) for the following factors: versatility (3.6 vs. 3.8), completeness of task (3.5 vs. 3.6), scope of action (3.4 vs. 3.8), social support (4.0 vs. 3.7), cooperation (3.6 vs. 3.4), qualitative work demands (2.2 vs. 2.1), works disruptions (2.7 vs. 2.4), information and participation (3.6 vs. 3.3), and benefits (2.9 vs. 2.4).”

5. We note you have included a table to which you do not refer in the text of your manuscript. Please ensure that you refer to Table 4 in your text; if accepted, production will need this reference to link the reader to the Table.

Answer: A reference to Table 4 is now included (line 221).

 “Table 4 shows a comparison of PrAs in our study population (from 2014) and the comparative study population (from 2000).”

Reviewers' comments:

Reviewer's Responses to Questions

Comments to the Author

1. Is the manuscript technically sound, and do the data support the conclusions?

Reviewer #1: Yes

2. Has the statistical analysis been performed appropriately and rigorously? 

Reviewer #1: Yes

3. Have the authors made all data underlying the findings in their manuscript fully available?

Reviewer #1: No

Answer: The data cannot be shared publicly because of ethical restrictions and data protection issues as our dataset includes potentially identifying information.

4. Is the manuscript presented in an intelligible fashion and written in standard English?

Reviewer #1: No

Answer: The manuscript was proofread by a certified medical translator. 

5. Review Comments to the Author

Reviewer #1: 

1. The manuscript is full of typography errors; punctuations.

Answer: The manuscript was proofread by a professional medical translator. 

2. Language is main problem

Answer: The manuscript was proofread by a professional medical translator.

3. Not consistent throughout the document

Answer: The manuscript was proofread by a professional medical translator. 

4. Don’t use abbreviation in the abstract part

Answer: This was corrected.

5. In the background part there are incomplete sentences

Answer: This was corrected.

6. In tables the decimal places should be consistent

Answer: This was corrected in Tables 1, 2,3 and 4.

7. In the table reporting missing value is not necessary

Answer: This was corrected in Tables 1 and 2.

6. PLOS authors have the option to publish the peer review history of their article (what does this mean?). If published, this will include your full peer review and any attached files.

Do you want your identity to be public for this peer review? For information about this choice, including consent withdrawal, please see our Privacy Policy.

Reviewer #1: No

Answer to reviewer comment concerning response rate

Reviewer comment: “Statistically how it can be generalized with around 30% non-response rate?

Answer: The argument in our sentence was incorrect. The total study had a response rate of 70% of practices. Within the practices, nearly all physicians and practice assistants participated indicating a high interest in the topic.

The text was revised to: It is a strength of our study that it was based on a data set with a large number of participants (550 PrAs). Also, prior analyses had shown that the practice network from which this sample was taken is representative for German primary care practices.

---

## [Decision Letter · Decision Letter 1]

4 Sep 2020

PONE-D-20-07803R1

Practice assistants´ perceived mental work load: A cross-sectional study with 550 German participants addressing work content, stressors, resources, and organizational structure

PLOS ONE

Dear Dr. Hoffmann,

Thank you for submitting your manuscript to PLOS ONE. After careful consideration, we feel that it has merit but does not fully meet PLOS ONE’s publication criteria as it currently stands. Therefore, we invite you to submit a revised version of the manuscript that addresses the points raised during the review process.

Our reviewer has positively valued the set of revisions you have done on the manuscript during the past round of revisions. However, some very minor changes and amendments need your attention. Please refer to both the comments appended below and the file attached by our reviewer. In case the revisions done in your resubmission were accurate and well-detailed, I will proceed to make an editorial decision without the need of a new round of reviews.

We look forward to receiving your revised manuscript.

Kind regards,

Sergio A. Useche, Ph.D.

Academic Editor

PLOS ONE

Reviewers' comments:

Reviewer's Responses to Questions

**Comments to the Author**

1. If the authors have adequately addressed your comments raised in a previous round of review and you feel that this manuscript is now acceptable for publication, you may indicate that here to bypass the “Comments to the Author” section, enter your conflict of interest statement in the “Confidential to Editor” section, and submit your "Accept" recommendation.

Reviewer #1: (No Response)

2. Is the manuscript technically sound, and do the data support the conclusions?

Reviewer #1: Yes

3. Has the statistical analysis been performed appropriately and rigorously? 

Reviewer #1: Yes

4. Have the authors made all data underlying the findings in their manuscript fully available?

Reviewer #1: Yes

5. Is the manuscript presented in an intelligible fashion and written in standard English?

Reviewer #1: No

6. Review Comments to the Author

Reviewer #1: 1. The manuscript has typography errors which are highlighted in the pdf file.

2. On the result section of the manuscript I have a great concern with the response rate which is 70.3%. Statistically this will have an impact on the power and it needs clear justification.

3. Table 1 and 2 some continuous variables are presented in the table as mean and median, which is not acceptable, please remove all continuous variables (age, work experiences, Number of Number of physicians in practice, and Number of PrAs in practice) from the table and narrate the findings.

4. With regard to missing data; it needs clear justification on, how did you manage missing data. As you described in table 1 and table 2 you have variable with missing values.

5. In the reference section: When you write references like reference number 6: “Wagner A, Rieger MA, Manser T, Sturm H, Hardt J, Martus P, et al. Healthcare professionals'perspectives on working conditions, leadership, and safety climate: a cross-sectional study. BMC Health Serv Res. 2019; 19: 53. doi: 10.1186/s12913-018-3862-7.” You write the journal name in abbreviated form, but on the other reference like reference number 10: “Theorell T, Hammarstrom A, Aronsson G, Traskman Bendz L, Grape T, Hogstedt C, et al. A systematic review including meta-analysis of work environment and depressive symptoms. BMC Public Health. 2015. doi: 10.1186/s12889-015-1954-4.” you write the journal in full. So, your references should be uniform and journal names better to be in full description.

7. PLOS authors have the option to publish the peer review history of their article (what does this mean?). If published, this will include your full peer review and any attached files.

Reviewer #1: No

---

## [Author Response · Author response to Decision Letter 1]

16 Sep 2020

Dear Dr. Useche,

Please find our revision and answers regarding the additional comments of reviewer 1. We hope that the revision made will adequately address the open points.

Best regards,

Jan Hoffmann

Reviewers´ comments: 

1. The manuscript has typography errors which are highlighted in the pdf file.

Answer: Thank you for highlighting the typography errors. We corrected them accordingly. 

2. On the result section of the manuscript I have a great concern with the response rate which is 70.3%. Statistically this will have an impact on the power and it needs clear justification.

Answer: We performed a power calculation using the software G-Power 3.1 for both group comparisons to show appropriate sample sizes. 

Line 176-178:

“Power calculations were performed using the software G-Power 3.1 to determine the appropriateness of sample sizes used in the group comparisons.” 

Line 206-208:

“The power calculation revealed that the sample sizes compared (n=550 versus n=8.121) were sufficient to achieve 80% power to detect small effect sizes of d = 0.12. In the case of greater differences, the power achieved was even higher.”

Line 226-228:

“The power calculation revealed that the sample sizes compared (n=550 versus n=324) were sufficient to achieve 80% power to detect small effect sizes of d = 0.2. In the case of greater differences, the power achieved was even higher.” 

Considering our study population, we also feel very comfortable with a response rate of 70%. Struggles to achieve high response rates in similar populations are reported elsewhere:

 Redaèlli M, Bassüner S, Teschner D, Stock S. Practice nurses can do more: Online surveys of VERAH-graduates and practice owners. Zeitschrift für Allgemeinmedizin. 2014; 90: 517–22. doi: 10.3238/zfa.2014.0517-0522.

Mergenthal K, Beyer M, Güthlin C, Gerlach FM. Evaluation des VERAH-Einsatzes in der Hausarztzentrierten Versorgung in Baden-Württemberg. Zeitschrift für Evidenz, Fortbildung und Qualität im Gesundheitswesen. 2013; 107:386–93. doi: 10.1016/j.zefq.2013.07.003 PMID: 24075680.

3. Table 1 and 2 some continuous variables are presented in the table as mean and median, which is not acceptable, please remove all continuous variables (age, work experiences, Number of Number of physicians in practice, and Number of PrAs in practice) from the table and narrate the findings.

Answer: We do not see a contradiction in mixing continuous variables and categorical variables in Tables 1 and 2. For clarification, we would like to maintain these variables in the table as has been done in other PLOS ONE publications on cross-sectional studies (e.g., Dubois J, Bill A-S, Pasquier J, Keberle S, Burnand B, Rodondi P-Y. Characteristics of complementary medicine therapists in Switzerland: A cross-sectional study. PLoS ONE. 2019; 14:e0224098. doi: 10.1371/journal.pone.0224098 PMID: 31644559)

4. With regard to missing data; it needs clear justification on, how did you manage missing data. As you described in table 1 and table 2 you have variable with missing values.

Answer: The management of missing data is described in line 161-162:

“Non-plausible values were recoded as missing values. Missing data were managed by reporting valid percentages only.”

In Table 1 and 2 variables with missing data are corrected. Now, only valid percentages are reported. For all variables, the number of analyzed answers is stated in brackets for clarification.

5. In the reference section: When you write references like reference number 6: “Wagner A, Rieger MA, Manser T, Sturm H, Hardt J, Martus P, et al. Healthcare professionals'perspectives on working conditions, leadership, and safety climate: a cross-sectional study. BMC Health Serv Res. 2019; 19: 53. doi: 10.1186/s12913-018-3862-7.” You write the journal name in abbreviated form, but on the other reference like reference number 10: “Theorell T, Hammarstrom A, Aronsson G, Traskman Bendz L, Grape T, Hogstedt C, et al. A systematic review including meta-analysis of work environment and depressive symptoms. BMC Public Health. 2015. doi: 10.1186/s12889-015-1954-4.” you write the journal in full. So, your references should be uniform and journal names better to be in full description.

Answer: 

Thank you for your comment. We checked all references to make sure that they are cited uniformly with full description of journal names.

---

## [Editor Report · Decision Letter 2]

18 Sep 2020

Practice assistants´ perceived mental workload: A cross-sectional study with 550 German participants addressing work content, stressors, resources, and organizational structure

PONE-D-20-07803R2

Dear Dr. Hoffmann,

We’re pleased to inform you that your manuscript has been judged scientifically suitable for publication and will be formally accepted for publication once it meets all outstanding technical requirements.

Kind regards,

Sergio A. Useche, Ph.D.

Academic Editor

PLOS ONE
---

## [Editor Report · Acceptance letter]

22 Sep 2020

PONE-D-20-07803R2 

Practice assistants´ perceived mental workload: A cross-sectional study with 550 German participants addressing work content, stressors, resources, and organizational structure 

Dear Dr. Hoffmann:

I'm pleased to inform you that your manuscript has been deemed suitable for publication in PLOS ONE. Congratulations! Your manuscript is now with our production department. 

Kind regards, 

on behalf of

Dr. Sergio A. Useche 

Academic Editor

PLOS ONE